# Evolving brain function and connectivity patterns during mentalizing in children and adults
Réka Borbás[1,3], Plamina Dimanova[1,3], Dennis Saikkonen[1], Elena Federici[1,2], Sofia Scatolin[1] & Nora M. Raschle [1,2] ✉

Mentalizing, the ability to infer others' thoughts and intentions, relies on a network of brain regions whose functional connectivity changes across development. While prior research has focused on adults, little is known about task-based functional connectivity in this network during development. We use fMRI to examine mentalizing-related activation and task-based connectivity in 181 participants (80 children aged 6-14; 101 adults aged 20-61). Analyses assess age-related changes in activation and connectivity, and test whether connectivity mediates the relationship between age and mentalizing ability across neurofunctional groups. Adults outperform children in mentalizing accuracy, though children show age-related improvements. Activation patterns are largely overlapping across age groups, involving core regions such as the temporoparietal junction, precuneus, and medial prefrontal cortex. Connectivity analyses reveal that children show stronger local (frontal-frontal, posterior-posterior) connections, with increasing long-range (frontal-posterior) connectivity with age. Adults exhibit a more integrated network, though connectivity declines with age. Connectivity strength follows a quadratic trajectory, peaking in early adulthood (~ 32 years). Importantly, connectivity mediates the age-mentalizing relationship in children, but not in adults. These findings suggest a shift from local to distributed mentalizing network connectivity across development, followed by age-related decline, shedding light on lifespan changes in social cognition.

Human sociality relies on a versatile set of skills essential for navigating social environments across the lifespan. Social competence requires the integration of multiple cognitive and affective processes, including executive function, memory, emotion processing, and emotion regulation[1,2]. Mentalizing, the ability to infer others' thoughts, feelings, and intentions while distinguishing them from one's own, is one of these key social skills[1,3]. Proficient mentalizing is associated with fewer behavioral problems[4], reduced somatic complaints[5], and better long-term psychological well-being[6]. Impairments in mentalizing are implicated in various psychiatric conditions, including depression, borderline personality disorder, alexithymia, and autism spectrum disorder[7–10].

Mentalizing develops alongside various cognitive and behavioral skills, including attention, language, memory, and executive functions[1,11]. Children typically acquire explicit mentalizing abilities between four and six years, as evidenced by their ability to verbally recognize false beliefs and differentiate between their own and others' mental states[11–13]. Although precursors to mentalizing emerge as early as 13-18 months[14,15], explicit and response-based forms of mentalizing that require verbal reasoning, perspective-taking, and cognitive control continue to develop throughout middle childhood and adolescence, and can be reliably assessed from approximately 5–6 years of age onward[12,13,16]. Accordingly, the present study targeted an age range (6-14 years) in which children can both understand task instructions and perform explicit theory of mind judgements with sufficient reliability. Furthermore, findings on mentalizing skills across the lifespan are mixed, with some adult studies reporting skill improvements while others suggest age-related declines for mid to late adulthood[17–19]. Recent evidence emphasizes the importance of understanding how age affects mentalizing, especially in distinguishing between cognitive and emotional components[20]. Furthermore, the need for more naturalistic and complex tasks is highlighted[21].

Two decades of neuroimaging research have identified a so-called mentalizing network, a set of brain regions consistently activated during mentalizing tasks. It is thought to support the ability to attribute thoughts, emotions, and intentions to oneself and others and includes medial

[1]Psychological Institute, Jacobs Center for Productive Youth Development, University of Zurich, Zurich, Switzerland. [2]Neuroscience Center Zurich, University of Zurich and ETH Zurich, Zurich, Switzerland. [3]These authors contributed equally: Réka Borbás, Plamina Dimanova. ✉e-mail: nora.raschle@jacobscenter.uzh.ch

prefrontal cortex (PFC), precuneus (PC), anterior and posterior cingulate cortices, bilateral temporoparietal junction (TPJ), and anterior temporal regions[22–25]. While meta-analyses mainly summarize findings in adults, we recently expanded on this body of knowledge by synthesizing functional neuroimaging studies in children and adolescents[23]. Accordingly, three key regions, the medial PFC, PC, and right TPJ, are consistently activated during mentalizing in children, adolescents, and adults. However, with increasing age, across adolescence and into young adulthood, additional areas are recruited, including bilateral inferior, middle, and superior frontal gyri, insula, and occipital pole. Developmental studies therefore indicate that the mentalizing network is functionally distinct in early childhood, though neural signaling linked to mentalizing seems to refine with age[4,26,27].

Mentalizing, like any other cognitive function, relies not only on the activation of individual brain regions but also on their coordinated coactivation[28,29]. Functional connectivity (FC) refers to temporally coordinated neural activity changes between brain regions of interest[30]. Understanding functional connectivity, in addition to task-related activation, thus provides deeper insight into the dynamic neural processes that support complex skills (i.e., mentalizing). Various studies show that both local activation differences and altered functional connectivity are associated with impairments in mentalizing, as observed in both clinical and non-clinical populations[31–34]. Increasing evidence thus implicates network dysfunction as a key factor in the pathogenesis of neuropsychiatric disorders[35].

Most developmental functional connectivity research has focused on resting-state assessments, which capture spontaneous intrinsic activity patterns. Such studies indicate potentially non-linear life course trajectories for the functional brain connectome, peaking around the fourth decade of life (i.e., around 38 years;[36]). The overall emergence, early development and subsequent decline of intrinsic functional connectivity has been linked to variations in human cognition and behavior, and is thus suggested to enable cognitive functioning[36,37].

At the system level, the rapid development of the resting-state default mode network (DMN; peaking around 32 years) is of particular relevance to the present investigation due to the networks' significant overlap with brain regions implicated in mentalizing[29,38,39]. While the DMN appears fragmented in younger children, it becomes more integrated and cohesive with age (e.g.,[36,38,40]). Notably, stronger long-range functional connections linking anatomically distant regions may be critical for the emergence of a mature DMN architecture and higher order cognition.

Although less common, task-based functional connectivity provides insights into the coordinated activation between relevant brain regions during specific cognitive tasks (i.e., mentalizing). Task-based connectivity has in fact been suggested to provide a more distinct and reliable predictor of specific behaviors compared to resting-state connectivity[41–43]. However, while task-based activation during mentalizing is well-documented[8,16,23,44,45], task-based functional connectivity remains less understood. Using seed-based functional connectivity, Burnett and Blakemore (2009) identified anterior prefrontal cortex connectivity with the TPJ, posterior superior temporal sulcus (STS), and anterior temporal cortex, all of which strengthened during mentalizing in both adolescents and adults[46]. Adolescents showed stronger anterior PFC-TPJ/posterior STS connectivity than adults, suggesting developmentally specific patterns for mentalizing network connectivity. Similarly, Richardson et al., (2018) used a movie-task paradigm to identify a mentalizing network in 3-12-year-olds, which resembles that of adults but becomes a more integrated and functionally distinct network with age[27]. Stronger intra-network connectivity and increased anti-correlation with the pain network suggested greater specialization over time.

Examining both regional activation and task-based functional connectivity dynamics across childhood and adolescence may, however, enhance our understanding of the neural mechanisms underlying complex skill acquisition, such as mentalizing. To close this gap in evidence, this study aims to investigate behavioral and whole-brain functional activation correlates of mentalizing, along with associated functional connectivity patterns, in typically developing children ($n = 80$; ages 6-14) and adults ($n = 101$; ages 20–61).

Based on prior meta-analytic[23] and task-based findings[16], we hypothesize that (**H1**) children and adults will perform above chance levels in a mentalizing task, with performance improving with age in children. (**H2**) Neural activation in key regions of the mentalizing network (e.g., left/right TPJ [L/RTPJ], PC, right STS [RSTS], dorsomedial PFC [DMPFC], middle medial PFC [MMPFC], and ventromedial PFC [VMPFC]) will be observed in both children and adults. (**H3**) Task-based functional connectivity between these regions will increase with age with adults exhibiting stronger and more widespread connectivity (e.g., long-range connections[27,29]). Consequently, overall mentalizing network strength will be higher for adults compared to children. Finally, data across all participants will be used to explore whether the relationship between age and mentalizing skills is mediated by changes in connectivity strength and whether these associations vary for specific neurofunctional phases. We differentiate between children ($\leq 14$ years), characterized by ongoing maturation and increasing functional segregation, young adults (20-32 years), representing the ascending-to-peak phase of network integration with relative stability, and middle-aged adults (33-61 years), undergoing gradual connectivity decline; in line with[36,40,47,48].

## Results
### Mentalizing (Behavior)
Across the 20 mentalizing trials, part of the CAToon fMRI mentalizing task[16], children solved an average of 81.90% correctly (M = 16.38, SD = 2.29), while adults achieved 85.51% accuracy (M = 17.10, SD = 1.54; Table 1). A Welch's $t$-test revealed significantly higher scores in adults compared to children, $t(131) = -2.40$, $p = 0.018$, $d = -0.38$. In children, mentalizing task performance (i.e., accuracy) significantly increased with age ($r(75) = 0.34$, $p = 0.002$; Supplementary Fig. 1), whereas no association with age was observed in adults ($r(95) = -0.03$, $p = 0.778$).

### fMRI: Task-based neural correlates of mentalizing
**Children.** One-sample $t$-tests revealed increased neural activation in mentalizing-related brain regions in line with[23,24], including PC extending into posterior cingulate, bilateral TPJs extending into bilateral middle and STS and temporal poles and bilateral fusiform gyri, orbitofrontal cortex and in medial prefrontal areas extending into anterior cingulate cortex. Additionally, activation increases in limbic regions, including amygdala, hippocampus, and insula were observed (Table 2, Fig. 1; FWE-thresholded data provided through NeuroVault: https://neurovault.org/collections/JQTZMGIX/images/1011920/). There were no significant effects of age observed for any of the identified clusters. Post-hoc non-parametric partial correlations between age and neural activity were tested in seven a priori regions of interest (ROIs) including L/RTPJ, PC, RSTS, DMPFC, MMPFC, and VMPFC. While uncorrected analyses showed associations between age and RSTS ($r(76) = 0.29$, $p = 0.009$), and RTPJ ($r(76) = 0.27$, $p = 0.015$), these associations did not pass multiple comparison correction (adjusted $p = 0.007$; 0.05/7).

**Adults.** One-sample $t$-tests revealed activation increases in mentalizing-related areas, including PC, posterior cingulate, bilateral TPJs, bilateral middle and STS and temporal poles, bilateral fusiform gyri, orbitofrontal cortex, and medial prefrontal areas along with anterior cingulate cortex. Furthermore, limbic areas were involved, including bilateral insula, hippocampus, and amygdala (Table 3, Fig. 1; FWE-thresholded data provided through NeuroVault: https://neurovault.org/collections/JQTZMGIX/images/1011917/). No significant age effects were observed for any of the identified whole-brain clusters. Post-hoc non-parametric partial correlations for the seven key mentalizing ROIs showed no significant age effects after multiple comparison correction (adjusted $p = 0.007$; 0.05/7), uncorrected analyses showed associations between age and DMPFC ($r(97) = -0.25$, $p = 0.012$) and LTPJ ($r(97) = -0.23$, $p = 0.025$), while all other ROIs had $p > 0.05$.

**Conjunction.** To identify regions commonly activated across both adults and children, a conjunction analysis was performed using the ImCalc

**Table 1 | Group characteristics of children and adults**

| | | Children (N = 80) [mean ± SD (n)] | Adults (N = 101) [mean ± SD (n)] |
|---|---|---|---|
| Age [Years] | | 10.1 ± 2.28 (80) | 38.65 ± 9.82 (101) |
| Sex (male/female) | | 51/29 | 30/71 |
| IQ | Non-verbal IQ | 109.06 ± 12.53 (80) | 106.97 ± 9.83 (38) |
| | Verbal IQ | 116.58 ± 16.67 (79) | 112.65 ± 13.04 (37) |
| | Total IQ | 112.88 ± 10.70 (80) | 109.41 ± 8.90 (38) |
| ISCED | ISCED | 5.59 ± 2.02 (80)[A] | 5.34 ± 2.05 (100) |
| Mental well-being | SDQ *total score* | 7.41 ± 5.38 (79) | N/A |
| | BSI *global severity index* | N/A | 50.73 ± 8.63 (74) |
| In-scanner performance | Overall Accuracy across all 30 trials (correct answers in %) | 82.15 (79) | 87.41 (99) |
| | Accuracy in mentalizing across 20 AT and CT trials (correct answers in %) | 81.90 (79) | 85.51 (99) |

Total IQ based on the average of verbal and performance IQ, when available; *ISCED* International Standard Classification of Education; *BSI* Brief Symptom Inventory (a T-score of 63 and above considered elevated); *SDQ* Strengths and Difficulties Questionnaire (a score of 17 or more considered elevated);. *N/A* does not apply; [A] = in kids ISCED is reported for their parents' ISCED, average of both parents or mother's score; *AT* affective Theory of Mind trials; *CT* cognitive Theory of Mind trials.

**Table 2 | Peak activation reports for mentalizing in children (n = 80)**

| Brain region | Hem. | T | p_FWE-corr | k | MNI X | y | z |
|---|---|---|---|---|---|---|---|
| precuneus, posterior cingulate | R/L | 15.61 | <0.001 | 4071 | 4 | -56 | 40 |
| temporoparietal junction, inf., mid. & sup. temporal gyrus, mid. & sup. temporal pole, angular gyrus, insula, amygdala, parahippocampal gyrus | R | 14.09 | <0.001 | 4494 | 48 | -54 | 20 |
| temporoparietal junction, inf., mid. & sup. temporal gyrus, mid. & sup. temporal pole, angular gyrus, insula, amygdala | L | 10.97 | <0.001 | 3916 | -50 | -58 | 20 |
| sup. medial frontal gyrus, anterior cingulate cortex | R/L | 9.30 | <0.001 | 1384 | 4 | 56 | 20 |
| medial orbitofrontal gyrus, gyrus rectus | R/L | 8.15 | <0.001 | 263 | 4 | 54 | -18 |
| inf., mid., sup. occipital gyrus | R | 7.89 | <0.001 | 118 | 28 | -96 | -8 |
| fusiform gyrus, inferior temporal gyrus | L | 7.23 | <0.001 | 81 | -38 | -42 | -18 |
| fusiform gyrus, inferior temporal gyrus | R | 7.12 | <0.001 | 136 | 40 | -44 | -22 |
| superior frontal gyrus | R | 6.96 | <0.001 | 87 | 16 | 38 | 56 |
| inf. frontal gyrus (pars triangularis & operculum) | R | 5.99 | 0.004 | 26 | 54 | 28 | -4 |
| inferior occipital gyrus | L | 5.87 | 0.013 | 11 | -20 | -100 | -8 |
| supplementary motor area, medial superior frontal gyrus | R | 5.87 | 0.002 | 43 | 10 | 26 | 64 |
| hippocampus, parahippocampal gyrus | R | 5.78 | 0.006 | 21 | 22 | -10 | -16 |
| medial superior frontal gyrus | L | 5.59 | 0.002 | 45 | -14 | 48 | 44 |
| inf. frontal gyrus (pars triangularis) | L | 5.51 | 0.004 | 29 | -54 | 24 | 8 |
| hippocampus, amygdala | L | 5.29 | 0.011 | 13 | -28 | -6 | -20 |

*Hem* hemisphere, *ACC* anterior cingulate cortex, *IFG* inferior frontal gyrus, *OFC* orbitofrontal cortex, *inf.* inferior, *mid.* middle, *sup.* superior, *L/R* left/right, *T*-scores, *k* cluster size and xyz co-ordinates of peak voxel according to Montreal Neurological Institute (MNI).

function in SPM. Commonly activated voxels included the bilateral TPJ, PC extending into the posterior cingulate cortex and paracentral lobule, occipital lobe, temporal regions (bilateral inferior, middle, and STS, bilateral fusiform gyrus, and bilateral temporal poles), medial PFC extending into the anterior cingulate cortex, orbitofrontal cortex, and limbic regions, including the bilateral insula and right hippocampus (FWE-thresholded data provided through NeuroVault: https://neurovault.org/collections/JQTZMGIX/images/1011921/).

**ROI-to-ROI Functional Connectivity**
**Children**. ROI-to-ROI functional connectivity analyses yielded significant positive connectivity during mentalizing in eight connections, including two frontal-frontal and six posterior-posterior connections,

but no frontal-posterior connections (Fig. 2, Table 4). Significant connectivity increases with age were observed for eight connections, comprising seven frontal-posterior and one posterior-posterior connection, but no frontal-frontal connections (Fig. 2, Table 5). Non-parametric partial correlations between age and overall mentalizing network connectivity strength yielded a significant positive association (*r*(76) = 0.31, *p* = 0.006).

**Adults**. In adults ROI-to-ROI analyses revealed significant positive connectivity during mentalizing in fifteen connections, including three frontal-frontal, five posterior-posterior, and seven frontal-posterior connections (Fig. 2, Table 4). Testing age effects in adults revealed significant connectivity decreases in eight connections, including three

**Fig. 1 | Neural correlates of mentalizing. A** Neural activation in children and adults during mentalizing. **B** Conjunction findings representing both groups displayed using whole-brain FWE-correction for multiple comparison testing at p < 0.05; x = 6, y = -47, z = 21.

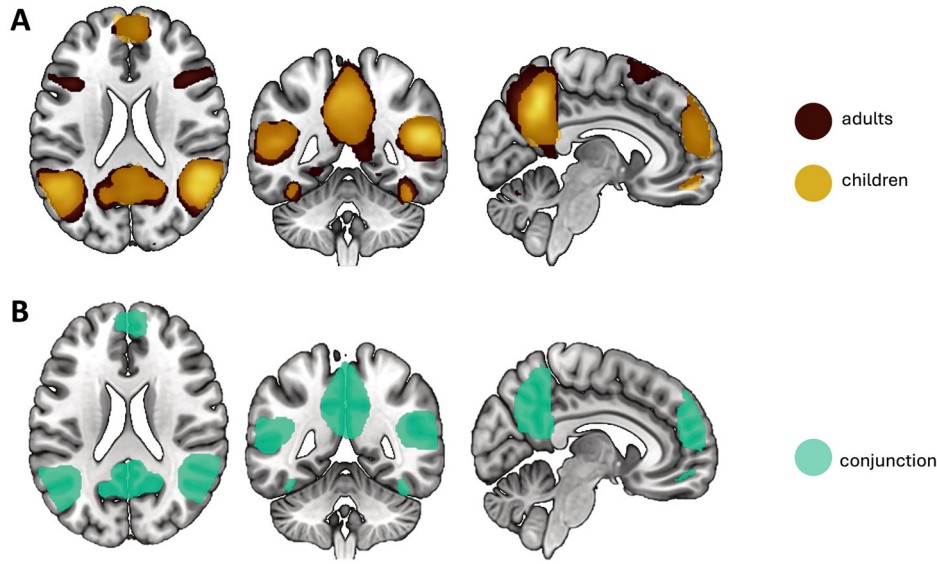

**Table 3 | Peak activation reports for mentalizing in adults (*n* = 101)**

| Brain region | Hem. | *T* | $p_{FWE\text{-}corr}$ | *k* | MNI X | y | z |
|---|---|---|---|---|---|---|---|
| temporoparietal junction, inf., mid. & sup. temporal gyrus, mid. & sup. temporal pole, angular gyrus, lingual gyrus, fusiform gyrus, inf. & mid. occipital gyrus, parahippocampal gyrus | R | 21.60 | <0.001 | 7597 | 46 | -60 | 16 |
| precuneus, posterior cingulate, cuneus | R/L | 21.42 | <0.001 | 6394 | 0 | -54 | 46 |
| temporoparietal junction, inf., mid. & sup. temporal gyrus, mid. & sup. temporal pole, supramarginal gyrus, angular gyrus, lingual gyrus, inf. & mid. occipital gyrus, insula, parahippocampal gyrus | L | 16.69 | <0.001 | 7611 | -44 | -56 | 18 |
| superior medial frontal gyrus, supplementary motor area, anterior cingulate cortex | R/L | 10.50 | <0.001 | 3004 | 6 | 52 | 26 |
| inf. frontal gyrus (pars triangularis & operculum), OFC | R | 10.45 | <0.001 | 843 | 56 | 26 | 6 |
| precentral gyrus, mid. frontal gyrus | L | 9.82 | <0.001 | 522 | -38 | 0 | 48 |
| medial orbitofrontal gyrus | R/L | 8.32 | <0.001 | 120 | 4 | 50 | -16 |
| mid. frontal gyrus, precentral | R | 7.09 | <0.001 | 306 | 40 | 4 | 46 |
| lingual gyrus | R | 6.53 | <0.001 | 50 | 18 | -56 | -6 |
| hippocampus, amygdala | R | 6.13 | 0.002 | 29 | 20 | -8 | -16 |
| parahippocampal gyrus, lingual gyrus, fusiform gyrus | R | 5.74 | 0.002 | 31 | 22 | -44 | -8 |
| hippocampus | R | 5.61 | 0.01 | 10 | 30 | -6 | -22 |

Cluster of k ≥ 10, surviving familywise error correction of *p* < 0.05 are reported. *Hem* hemisphere, *ACC* anterior cingulate cortex, *IFG* inferior frontal gyrus, *OFC* orbitofrontal cortex, *inf.* inferior, *mid.* middle, *sup.* superior, *L/R* left/right, T-scores, *k* cluster size and *xyz* co-ordinates of peak voxel according to Montreal Neurological Institute (MNI).

frontal-frontal, five frontal-posterior, but no posterior-posterior connections (Fig. 2, Table 5). Non-parametric partial correlations between age and overall mentalizing network connectivity strength showed a negative association ($r(97) = -0.22$, $p = 0.026$).

**Linear and Non-Linear Age Effects in Children and Adults**

**Age and Mentalizing Task Performance.** In the linear model, age was a significant predictor ($b = 0.02$, $t = 2.17$, $p = 0.032$), explaining 4.44% of the variance (adj. $R^2 = 0.04$, $F(3,174) = 3.74$, $p = 0.012$). Adding a quadratic term did not significantly improve model fit ($p = 0.063$), although it increased explained variance to 5.80% (adj. $R^2 = 0.06$, $F(4,173) = 3.73$, $p = 0.006$).

**Age and ROI-to-ROI Connectivity.** Multiple regression analyses tested the effect of age on overall connectivity strength during mentalizing trials. In the linear model, age was a significant predictor ($b = 0.002$, $t = 3.22$, $p = 0.002$), explaining 7.98% of the variance (adj. $R^2 = 0.08$, $F(3,177) = 6.20$, $p < 0.001$). Including a quadratic term significantly

improved model fit ($p < 0.001$) and increased explained variance to 17.15% (adj. $R^2 = 0.17$, $F(4,176) = 10.32$, $p < 0.001$). Both the linear ($b = 0.02$, $t = 5.08$, $p < 0.001$) and quadratic ($b = -0.000$, $t = -4.54$, $p < 0.001$) terms were significant. Calculating the apex of the quadratic curve showed that overall mentalizing network connectivity increases with age, peaking around 32 years before declining (see Fig. 3A).

**Moderated mediation.** Age ($b = 0.03$, $t(173) = 3.88$, $p < 0.001$) and group ($b = 0.19$, $t(173) = 3.10$, $p = 0.002$) significantly predicted long-range connectivity strength, with a significant age × group interaction ($b = -0.01$, $t(173) = -4.54$, $p < 0.001$; Fig. 3B; Table 6). Long-range connectivity strength increased with age in children ($b = 0.02$, $t(173) = 3.36$, $p = 0.001$) and in young adults ($b = 0.01$, $t(173) = 2.08$, $p = 0.039$), and was not significant predictor in middle-aged adults. Mentalizing task performance was positively predicted by long-range connectivity strength ($b = 6.79$, $t(172) = 3.65$, $p < 0.001$) and the significant interaction between long-range connectivity strength and group ($b = -2.65$, $t(172) = -3.01$, $p = 0.003$) adding significant explanatory power to the model (long-

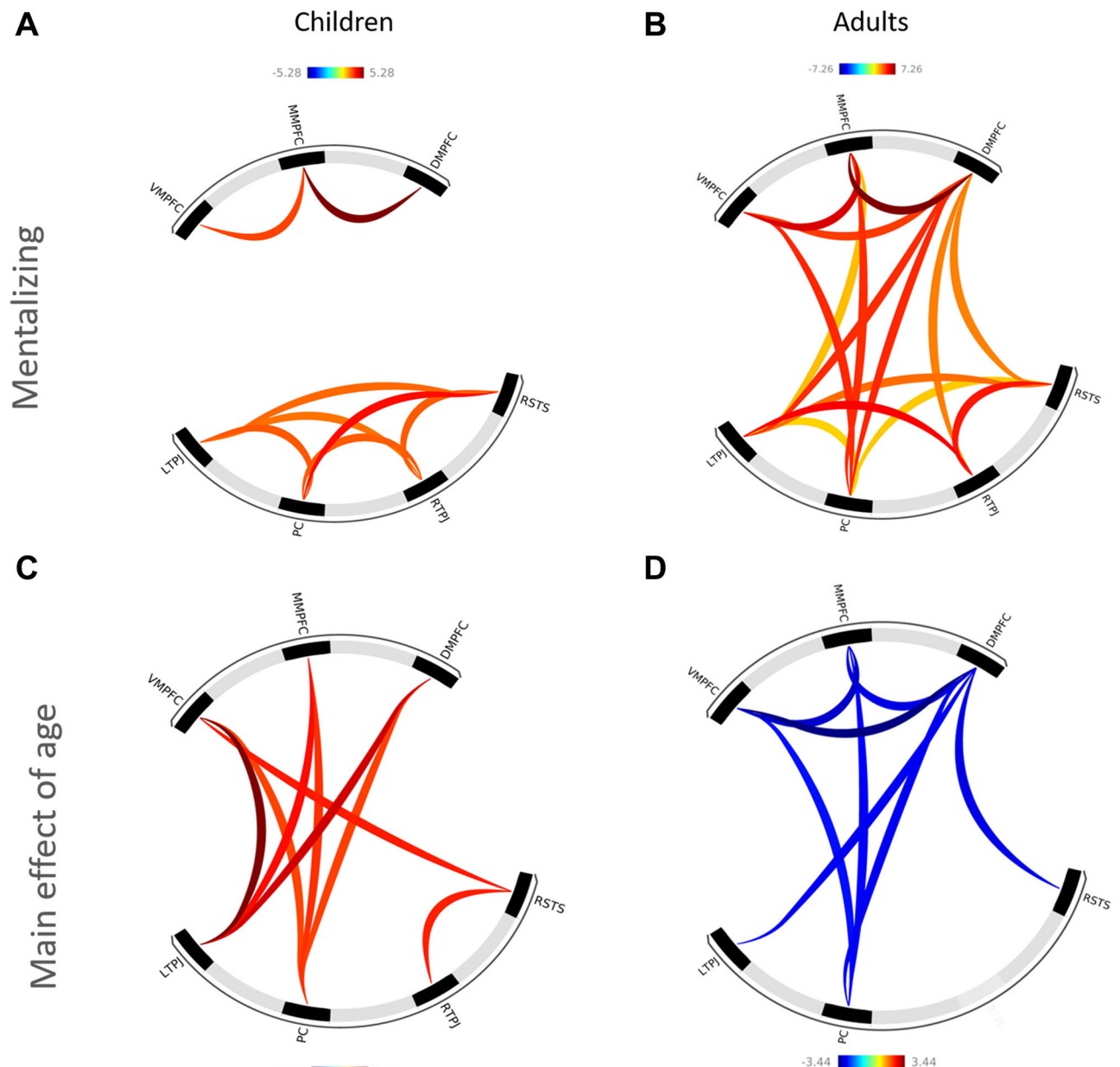

**Fig. 2 | Ring plots showing task-based ROI-to-ROI functional connectivity during mentalizing in children and adults.** Significant functional connectivity during mentalizing, accounting for age, sex, and site in (**A**) children and (**B**) adults. Main effect of age on connectivity strength, controlling for sex and site in (**C**) children and (**D**) adults. Color scale indicates the strength and direction of functional connectivity effects (*t*-values). Warm colors (yellow-red) denote positive effects, whereas cool colors (blue) denote negative effects. Color intensity reflects the magnitude of the effect. All analyses were false discovery rate corrected (p-FDR < 0.05) for multiple comparisons.

range connectivity strength x group: $F(1,172) = 9.04$, $p = 0.003$, $R^2$-change = 0.05). Within the individual age groups, mentalizing task performance was positively predicted by long-range connectivity strength only in children ($b = 4.13$, $t(172) = 3.66$, $p < 0.001$), while the effect was non-significant in young adults and negative but non-significant in middle-aged adults. The indirect effect of age on mentalizing abilities via long-range connectivity strength was significant only in children ($b = 0.07$, *BootSE* = 0.04, *95% BootCI [0.004, 0.158]*) and non-significant in young and middle-aged adults.

## Discussion

This study investigated mentalizing performance, its neural correlates, and task-based functional connectivity in children and adults, addressing gaps left by prior research that primarily focused on adults or resting-state

connectivity. While both children and adults engaged a largely overlapping core mentalizing network, connectivity patterns varied significantly with age. Behaviorally, adults outperformed children in mentalizing accuracy, though children showed age-related improvements. Neural activation analyses confirmed engagement of key mentalizing regions, including the temporoparietal junction (TPJ), precuneus (PC), and medial prefrontal cortex (PFC), across all participants (aligning with[23,24]). However, functional connectivity analyses revealed distinct developmental associations: Children exhibited strong local (frontal-frontal and posterior-posterior) connectivity, with increasing age linked to the emergence of long-range (frontal-posterior) connectivity. In contrast, adults displayed a more integrated network, but connectivity strength for long-range connections declined with age. Across all participants a quadratic pattern for overall network connectivity strength was observed peaking in early adulthood ( ~ 32 years).

**Table 4 | ROI-to-ROI connectivity in children and adults during mentalizing**

| Connection type | Connection ROI – ROI | Children (N = 80) | | Adults (N = 101) | |
|---|---|---|---|---|---|
| | | T (76) | p-FDR | T (97) | p-FDR |
| Frontal-frontal | DMPFC-MMPFC | 5.28*** | <0.001 | 7.26*** | <0.001 |
| | DMPFC-VMPFC | 1.46 | 0.220 | 4.61*** | <0.001 |
| | MMPFC-VMPFC | 3.26** | 0.007 | 5.98*** | <0.001 |
| Posterior-posterior | LTPJ-PC | 3.06* | 0.014 | 2.41* | 0.025 |
| | LTPJ-RSTS | 2.87* | 0.033 | 3.92*** | 0.001 |
| | LTPJ-RTPJ | 2.75* | 0.021 | 5.53*** | <0.001 |
| | RSTS-PC | 3.85** | 0.002 | 2.56* | 0.018 |
| | RSTS-RTPJ | 3.02* | 0.014 | 5.12*** | <0.001 |
| | PC-RTPJ | 3.01* | 0.014 | 1.52 | 0.154 |
| Frontal-posterior | DMPFC-LTPJ | -1.03 | 0.338 | 4.82*** | <0.001 |
| | DMPFC-PC | -0.04 | 0.875 | 4.87*** | <0.001 |
| | DMPFC-RSTS | 0.18 | 0.875 | 3.58** | 0.001 |
| | DMPFC-RTPJ | 0.26 | 0.891 | 3.61** | 0.001 |
| | MMPFC-LTPJ | -1.22 | 0.220 | 2.72* | 0.012 |
| | MMPFC-PC | 0.54 | 0.875 | 4.81*** | <0.001 |
| | MMPFC-RSTS | 0.27 | 0.875 | 2.05 | 0.057 |
| | MMPFC-RTPJ | 1.02 | 0.744 | 1.10 | 0.287 |
| | VMPFC-LTPJ | -1.73 | 0.107 | 1.90 | 0.075 |
| | VMPFC-PC | 0.96 | 0.752 | 4.86*** | <0.001 |
| | VMPFC-RSTS | -0.35 | 0.875 | 1.34 | 0.202 |
| | VMPFC-RTPJ | -0.07 | 0.875 | 0.96 | 0.339 |

*DMPFC* dorsomedial prefrontal cortex; *LTPJ* left temporoparietal junction; *MMPFC* middle medial prefrontal cortex; *PC* precuneus; *RSTS* right superior temporal sulcus; *RTPJ* right temporoparietal junction; *VMPFC* ventromedial prefrontal cortex; Connection type: the order of the ROIs indicates no directionality, all connections are symmetrical; *p < 0.05, **p < 0.01, ***p < 0.001.

**Table 5 | Main effect of age on ROI-to-ROI connectivity in children and adults during mentalizing**

| Connection type | Connection ROI – ROI | Children (N = 80) | | Adults (N = 101) | |
|---|---|---|---|---|---|
| | | T (76) | p-FDR$_{one-tailed}$ | T (97) | p-FDR |
| Frontal-frontal | DMPFC-MMPFC | 0.66 | 0.307 | -2.73* | 0.033 |
| | DMPFC-VMPFC | 1.75 | 0.098 | -3.44* | 0.018 |
| | MMPFC-VMPFC | 1.63 | 0.103 | -2.94* | 0.033 |
| Posterior-posterior | LTPJ-PC | 0.79 | 0.284 | 0.20 | 0.930 |
| | LTPJ-RSTS | 1.49 | 0.124 | -0.88 | 0.535 |
| | LTPJ-RTPJ | 0.93 | 0.262 | 0.07 | 0.993 |
| | RSTS-PC | -0.16 | 0.138 | -1.40 | 0.367 |
| | RSTS-RTPJ | 2.56* | 0.026 | -1.28 | 0.367 |
| | PC-RTPJ | -0.47 | 0.355 | 0.31 | 0.884 |
| Frontal-posterior | DMPFC-LTPJ | 3.14* | 0.013 | -2.66* | 0.033 |
| | DMPFC-PC | 2.34* | 0.031 | -2.65* | 0.033 |
| | DMPFC-RSTS | 1.67 | 0.103 | -2.75* | 0.033 |
| | DMPFC-RTPJ | 0.89 | 0.262 | -1.07 | 0.433 |
| | MMPFC-LTPJ | 2.73* | 0.026 | -1.29 | 0.367 |
| | MMPFC-PC | 2.45* | 0.029 | -2.57* | 0.035 |
| | MMPFC-RSTS | 1.38 | 0.138 | -1.26 | 0.367 |
| | MMPFC-RTPJ | -0.63 | 0.307 | 0.58 | 0.695 |
| | VMPFC-LTPJ | 3.75** | 0.004 | -1.09 | 0.433 |
| | VMPFC-PC | 2.31* | 0.031 | -2.48* | 0.039 |
| | VMPFC-RSTS | 2.59* | 0.026 | -0.69 | 0.643 |
| | VMPFC-RTPJ | 0.26 | 0.418 | 0.01 | 0.993 |

*DMPFC* dorsomedial prefrontal cortex; *LTPJ* left temporoparietal junction; *MMPFC* middle medial prefrontal cortex; *PC* precuneus; *RSTS* right superior temporal sulcus; *RTPJ* right temporoparietal junction; *VMPFC* ventromedial prefrontal cortex; Connection type: the order of the ROIs indicates no directionality, all connections are symmetrical; *p < 0.05, **p < 0.01, ***p < 0.001.

Moderated mediation analyses further showed that long-range connectivity strength mediated the association between age and mentalizing performance in children ($\leq 14$ years) but not in younger (23-32 years) or middle-aged adults (33-61 years), highlighting the importance of network integration for social cognition in early development.

Whole-brain group analyses, along with a conjunction analysis of shared activation, confirmed the engagement of a similar neural mentalizing network for children and adults, including the medial PFC, posterior cingulate cortex, PC, TPJ, temporal poles and gyri, and limbic areas. These findings align with previous research suggesting that some of the core neural correlates of mentalizing emerge early in childhood and remain comparable in adulthood[16,23,27]. In children and adults, activation was observed across these key regions, with no significant within-group age effects. Prior studies have reported age-related differences in prefrontal mentalizing regions, with findings of both increased and decreased activation, depending on task demands, cognitive processing requirements, and age range studied[8,23,24,49–51]. The largely overlapping mentalizing network observed here supports the notion of a relatively stable core set of mentalizing correlates from middle childhood onward. However, future studies including younger children and older adults are needed to further clarify life course development. Furthermore, more adaptive tasks could be tested in wider

age range and potentially reveal greater age-related variability in task-evoked activation[52].

Children exhibited significant task-based local connectivity when assessing an a priori defined, ROI-to-ROI-based, functional mentalizing network, with strong frontal-frontal and posterior-posterior connections but no frontal-posterior connectivity. This observation aligns with previous evidence investigating the DMN and suggests a more fragmented and disconnected functional connectome in younger children[38]. As age increased, seven long-range connections emerged across childhood and adolescence, reflecting greater coactivation of frontal and parietal regions, along with one more posterior-posterior connection (right TPJ-right STS). Our findings align with the proposed shift from local to more integrated functional network organization[29,48], likely reflecting underlying neurodevelopmental processes such as myelination and white matter tract maturation, which continue into young adulthood[29,48,53].

In developmental populations, research on functional connectivity during mentalizing tasks remains limited. However, our findings align with past studies that have identified increasing network specialization with age, as indicated by changes in functional connectivity. For example, past studies similarly found that adolescents showed enhanced ventromedial

**Fig. 3 | Effect of age on connectivity strength and mentalizing performance. A** Quadratic association between age (in years; x-axis) and overall mentalizing network connectivity (y-axis; Fisher-z transformed correlational scores) demonstrated an inverse u-shaped trajectory, peaking around 32 years of age. **B** Long-range connectivity strength varied by age and group, increasing in children and showing a weaker but still significant increase in young adults, with no significant age association in middle-aged adults. Across groups, mentalizing task performance was positively associated with long-range connectivity strength in children. In the moderated mediation model, a significant indirect effect of age on mentalizing performance via long-range connectivity strength was observed for the child group (see also, Table 6).

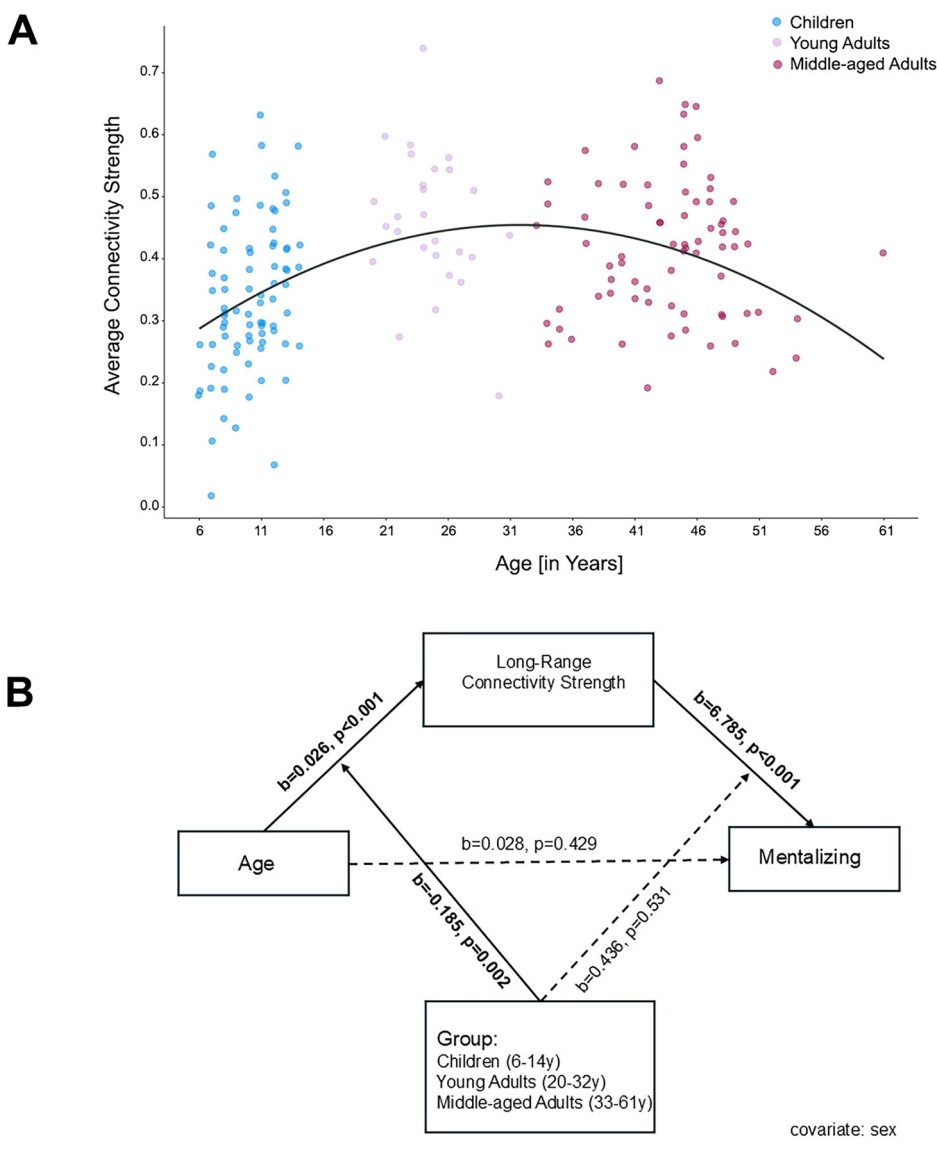

prefrontal–temporoparietal connectivity[46], while children (3-12 years) exhibited age-related increases in connectivity between the temporoparietal junction, precuneus, and medial prefrontal cortex[27]. Notably, some studies report stable connectivity patterns in mentalizing regions across development[54,55]. Such differences may stem from variations in task design, sample characteristics, and analytical methods used (e.g., psychophysiological interaction analyses[46] or[55] vs. whole-brain functional connectivity as in[27] and the present task-based connectivity findings).

Children exhibited predominantly local connectivity, with long-range connections emerging gradually with age. In contrast, adults showed a more integrated mentalizing network, characterized by significant frontal-frontal, posterior-posterior, and frontal-posterior task-based connectivity, reflecting a mature balance between local and global network integration. However, increasing age in adults was linked to a decline in overall network connectivity strength, particularly in frontal-frontal and frontal-posterior connections, while posterior-posterior connectivity remained stable. The finding of decreasing connectivity strength with increasing age in adults aligns with previous work by[36,47,56]. For example, Siman-Tov and colleagues (2017) found decreasing connectivity strength when comparing participants in their mid-adulthood (41-61 years) to younger adults (21-40 years)[47]. Furthermore, findings from Fehlbaum and colleagues (2021; employing meta-analytic connectivity modeling) align with our observations by highlighting coactivations of the superior/middle temporal regions,

precuneus, medial superior frontal gyri, and middle/inferior and middle/medial frontal gyri, with additional links to the thalamus, basal ganglia, and inferior/superior parietal lobule during mentalizing[23].

Notably, we observed asymmetric age-related effects within temporoparietal and superior temporal regions, consistent with prior evidence[23,57]. Specifically, the left TPJ and right STS exhibited distinct developmental trajectories. Such hemispheric differentiation may reflect a shift from perceptually anchored to more abstract inferential processes in mentalizing. The left TPJ has been linked to conceptual perspective-taking and language-mediated reasoning about others' beliefs, whereas the right TPJ and right STS, although not both showing age effects in the present study, are generally implicated in visual-social perception and the decoding of dynamic social cues (such as gaze direction, motion, and facial expressions;[58–60]). Developmental refinement within these regions may therefore mirror the transition from perceptual social understanding in childhood toward increasingly abstract reasoning and linguistic forms of mental state attribution in adolescence and adulthood. This interpretation aligns with broader models of cortical specialization suggesting that functional differentiation increases during brain maturation before later convergence or reduced segregation in adulthood[40]. However, as the left STS was not included among our ROIs and whole-brain activation appeared largely bilateral, any lateralization effects should be interpreted cautiously.

**Table 6 | Summary of Moderated Mediation Analysis testing whether the relation between age and mentalizing is mediated by long-range connectivity strength (Conn$_{LR}$) and whether such associations differ by group (children, young adults, middle-aged adults)**

| Predictor | b | t | p | F (df1, df2) | R²-change |
|---|---|---|---|---|---|
| LR-Conn$_{LR}$ | | | | 14.44 (4,173) | 0.25 |
| Age | 0.03 | 3.88 | <0.001 | | |
| Group | 0.19 | 3.10 | 0.002 | | |
| Age × Group | -0.01 | -4.54 | <0.001 | 20.63 (1,173) | 0.09 |
| Mentalizing | | | | 4.85 (5,172) | 0.12 |
| Con$_{LR}$ | 6.79 | 3.65 | <0.001 | | |
| Age | 0.03 | 0.79 | 0.429 | | |
| Group | 0.44 | 0.63 | 0.531 | | |
| LR-ConnS × Group | -2.65 | -3.01 | 0.003 | 9.04 (1,172) | 0.05 |
| **Conditional Effects of Age and Connectivity Strength by Group** | | | | | |
| Group | b (Age → Conn$_{LR}$) | t | p | b (Conn$_{LR}$ → Mentalizing) | t | p |
| Children | 0.02 | 3.36 | 0.001 | 4.13 | 3.66 | <0.001 |
| Young Adults | 0.01 | 2.08 | 0.039 | 1.48 | 1.83 | 0.069 |
| Middle-Aged Adults | -0.002 | -0.52 | 0.601 | -1.17 | -0.93 | 0.354 |
| **Indirect Effects of Age on Mentalizing Task Performance** | | | | | |
| Group | Effect | BootSE | BootLLCI | BootULCI | |
| Children | 0.07 | 0.04 | 0.004 | 0.158 | |
| Young Adults | 0.01 | 0.01 | -0.006 | 0.034 | |
| Middle-Aged Adults | 0.002 | 0.01 | -0.011 | 0.015 | |

*Conn$_{LR}$*Long-Range Connectivity Strength.

Across all participants age was a significant predictor of mentalizing task performance, though a quadratic model did not substantially improve the fit over a linear model. In contrast, overall connectivity strength tested in our full sample followed a quadratic trajectory, increasing with age, peaking around age 32 before declining. These changes in brain connectivity have been postulated as a fundamental mechanism underlying cognitive development. As functional and structural networks mature, the development of higher-order cognitive abilities is enabled, including for example theory of mind skills[2]. The observed pattern of early increases followed by decreases in network strength aligns with evidence reporting a quadratic maturational trajectory for the DMN. For example, large-scale evidence by Sun and colleagues (2025), reflecting on resting-state fMRI data from over 33,000 individuals across the lifespan identified a similar nonlinear increase in connectome strength followed by a decline[36]. The evidenced trajectory was primarily driven by changes in middle- and long-range connections, echoing our task-based findings. Notably, the rapidly increasing trajectory, followed by declines, seems specific to the DMN including many regions linked to mentalizing, but is absent in other networks, such as the sensorimotor network[40,47,56].

Given the observed age-related patterns in mentalizing performance and long-range connectivity (rising in childhood and peaking in early adulthood), we conducted a moderated mediation analysis investigating whether changes in the long-range connectivity strength mediate the relationship between age and mentalizing skills across three neurofunctional groups (children, young adults, middle-aged adults; in line with[36,40,47,61]). We found that age and group significantly predicted long-range connectivity strength, with age-related increases in children, a weaker but still positive effect in young adults, and no significant association in middle-aged adults. Long-range connectivity strength significantly mediated the relationship between age and mentalizing task performance only in children, indicating that the emergence of long-range connectivity plays a central role in supporting social cognitive skills during this developmental time window. Long-range connectivity strength continued with a significant, although weaker positive association in young adults, but no significant association

with mentalizing performance in middle-aged adults. Interestingly, despite age-related declines in overall connectivity strength, mentalizing performance did not decline with age across adulthood in our sample. This pattern suggests a possible shift in the neural mechanisms supporting mentalizing across the lifespan, or the influence of compensatory mechanisms in middle-aged adults. Contrary to our findings, past work in adults has identified age-related declines in cognitive performance. Moran et al.[62] reported reduced medial PFC involvement in older adults (~70+ years) during social cognitive tasks, accompanied by declines in mental state attribution abilities, underscoring the importance of using paradigms with varying task demands and encompassing the full lifespan in future research.

While the neural architecture supporting social cognition undergoes reorganization across the lifespan, our findings may point towards a recalibration of neural mechanisms rather than a decline in function, with decreases in connectivity strength not necessarily implying diminished social cognitive abilities. In adulthood, mentalizing skills may rely on more stable or compensatory mechanisms beyond connectivity strength alone. For example, research indicates that older adults may engage alternative neural pathways or networks to maintain cognitive functions, including social cognition, despite age-related declines in certain brain regions[63,64]. In our study, the observed early stabilization of functional neural activation across children and adults warrants critical interpretation, more challenging tasks could potentially reveal greater age-related variability in task-evoked activation (e.g.,[52]) or skill-related changes. In children and adolescents, mentalizing performance was positively associated with long-range connectivity strength. The indirect effect of age on mentalizing performance via long-range connectivity strength was significant exclusively in this group, supporting the notion that network integration plays a crucial role in shaping social cognitive abilities during early development[29,36,40,48,56].

The present study's age-related findings are based on a cross-sectional sample, limiting causal developmental inferences. Future research with prospective longitudinal data is needed to replicate these network changes over time. Additionally, our sample does not cover all developmental periods (in particular infants, toddlers, adolescents between 14 and 20 years and

older adults ( > 61 years) are missing), which precludes drawing conclusions about continuous lifespan trajectories. As our task was designed for explicit, response-based mentalizing, children younger than 6 years were not included. Future work should adapt paradigms for younger participants to capture the emergence of early implicit and explicit mentalizing abilities during the preschool years (e.g., through more adaptive tasks;[52] The age distribution within our sample and age groups was non-normal, which may have inflated standard errors and reduced statistical power, limiting our ability to detect age-related effects in underrepresented groups. As our sample does not include participants between 14 and 20 years, a key age window for pubertal and social-cognitive development during adolescence and emerging adulthood remains beyond the scope of our examination. The grouping based on neurofunctional phases ( ≤ 14, 20-32, ≥33 years) was therefore data-driven, reflecting the observed connectivity trajectory rather than predefined developmental stages, but this age gap persists an important limitation for future work. Unequal sex distribution may have also influenced the results; while sex effects were controlled for, larger and more balanced samples are necessary to explore sex-specific functional connectivity trajectories in greater detail (e.g.[45,65]). This study does not directly link observed network changes to real-life social abilities, which is a crucial step for understanding their broader implications. Although ComBat-harmonization was applied to minimize site-related variance, subtle scanner-specific effects cannot be entirely ruled out, underscoring the importance of replication in independent multi-site samples. Future research should clarify the directionality of connectivity changes (e.g., strengthening of positive correlations or loss of anticorrelations;[40]) and integrate task-based measures with resting-state or dynamic stimulus-driven data[31,66].

## Methods

### Participants

Participants were recruited via social media, flyers, online platforms, and word-of-mouth and tested at two sites in Switzerland (Basel and Zurich). 80 children (29 females, 51 males; mean age = 10.1 years, range: 6-14 years) and 101 adults were included in the analyses (71 female, 30 males; mean age = 38.65 years, range: 20-61 years). 27 adults and 33 children were previously part of[16]. Participants had sufficient German proficiency to understand the fMRI task instructions and an average or above-average IQ (Table 1). 14 children and 14 adults were excluded prior to analyses start due to low data quality, braces, excessive movement, claustrophobia, incomplete tasks, developmental delay, or visual impairments. This study was approved by the Ethics Committee Northwest and Central Switzerland (Basel) and the Cantonal Ethics Committee Zurich. Adult participants provided written consent, while parental written consent and child assent were obtained for children. All ethical regulations relevant to human research participants were followed.

### Behavioral assessments

**Intelligence/Education.** IQ is reported using the German adaptation of the Wechsler Intelligence Scale for Children (WISC-IV;[67]) and the Wechsler Adult Intelligence Scale for adults (WAIS-IV;[68]) using standardized norm scores for matrix reasoning (non-verbal IQ) and vocabulary (verbal IQ; Table 1). Since IQ scores were only available for some adults, we additionally calculated their highest level of education, assessed using the International Standard Classification of Education[69]. The average educational attainment was ISCED level 5.34 ± 2.05, indicating that most had completed tertiary education levels.

**Mental well-being.** Parent-report for the Strengths and Difficulties Questionnaire (SDQ[70]) were used to assess children's psychological adjustment. The domains conduct problems, emotional symptoms, hyperactivity and peer relationship problems were summed to yield the total difficulties score. Adults completed the Brief Symptom Inventory (BSI), a 53-item self-report questionnaire evaluating mental well-being[71]. A global severity index (GSI) reflecting psychological well-being was

calculated by averaging the scores of all 53 items, then converted to a standardized T-score adjusted for sex and age. None of the adults or children met clinically relevant sum scores (GSI ≥ 63 or SDQ total ≥17).

### Neuroimaging task

**fMRI.** During neuroimage acquisition each participant completed CAToon, an open-source fMRI mentalizing task including a preparatory training outside of the scanner (see[16]). CAToon consists of 30 cartoon stories, with 10 stories in each of three conditions: two experimental conditions (forming mentalizing: affective theory of mind, and cognitive theory of mind conditions), and one control condition (physical causality). Participants viewed incomplete picture stories with three consecutive images, each displayed for three seconds. They were then prompted to complete the story by selecting one of three possible endings shown simultaneously on the screen. During training, participants solved three example trials and were instructed to make decisions by considering the characters' thoughts and feelings or by applying basic physical rules (control condition, e.g., ice melting, objects falling). Participants indicated their choice of the most likely story ending using a button box, pressing one of three buttons within a 7-10 second answer window (task details in Supplementary Fig. 2 and Supplementary Method 1).

**Mentalizing (Behavior).** In-scanner performance was assessed by calculating overall accuracy as the percentage of correct answers across experimental and control conditions (i.e., mentalizing and physical causality) and accuracy in mentalizing trials as the percentage of correct answers across the experimental conditions only (Table 1). In-scanner task performance data in one child and two adults were missing due to a technical malfunction of the button box. To ensure comprehension and task reliability, all participants completed an extensive training session prior to scanning, during which they verbally explained their reasoning for each trial, confirming an understanding of the task demands. Moreover, visual inspection of first-level activation maps for these participants confirmed expected task-related activation patterns, supporting the reliability of the group-level findings.

**Neuroimage acquisition.** Whole-brain T2*-weighted echo-planar images were collected on a Siemens 3T Prisma MR scanner using a 20-channel head coil in Basel and a 3T GE SIGNA Premier scanner using a 48-channel head coil in Zurich. At both sites dummy scans were included and later discarded accounting for T1-equilibration effects. Structural mprage (in Basel) and BRAVO (in Zurich) T1-weighted images were used for co-registration (acquisition details in Supplementary Method 2).

### Neuroimage analyses

**Regions of interest (ROIs).** ROIs were functionally defined based on activation maps from an independent group analysis of 462 neurotypical adults[72] and comprised bilateral TPJ [L/RTPJ], PC, right STS [RSTS], dorsomedial PFC [DMPFC], middle medial PFC [MMPFC], and ventromedial PFC [VMPFC]. Anatomical labels are provided for orientation only. All ROIs are visualized in Supplementary Fig. 3.

**fMRI preprocessing.** Analyses were conducted in SPM ([73]; RRID:SCR-007037; release 12.7771; in MATLAB (R2023a)). Functional images were realigned to the first image, co-registered to each participant's structural image, normalized to standard space (Montreal Neurological Institute template) and smoothed using an 8 mm full width at half maximum isotropic kernel. Movement and signal outliers were detected using the ART toolbox[74]. ART identifies outlier volumes based on volume-to-volume displacement derived from the six realignment parameters. Volumes showing translational change exceeding 2 mm or global intensity outliers were modeled with single-impulse nuisance ('spike') regressors in the first-level GLM, rather than deleted or temporally interpolated (see Supplementary Method 3). Six motion parameters and one aggregated outlier regressor were added in each participant's first-

https://doi.org/10.1038/s42003-026-09562-6 **Article**

level model. Given the block design and low fraction of censored volumes, this approach was expected to have negligible impact on model efficiency. High-pass filtering of 0.01 Hz (128 s) was applied. Activation maps were labeled using the automated anatomical labeling atlas[75].

**fMRI - Random effects.** First-level analyses modeled task events as boxcar functions convolved with the canonical hemodynamic response function. Regressors of interest included the full trial durations for "affective theory of mind", "cognitive theory of mind", and "physical causality" conditions, including decision time (in line with[16], Supplementary Fig. 2). Mentalizing was assessed by contrasting "affective theory of mind" and "cognitive theory of mind" trials to the control condition ("physical causality"). Whole-brain analyses (*n-children* = 80, *n-adults* = 101) included age, site, and sex as covariates of no interest. In a subsequent step, the effects of age were further explored for significant clusters. Results are presented with whole-brain familywise error (FWE) correction at p < 0.05.

**Functional connectivity.** For functional connectivity, the CONN toolbox (release 22.a;[76]) was used. Anatomical data was normalized into standard MNI space and resampled to 1 mm isotropic voxels using SPM unified segmentation and normalization algorithm[77,78] with the default IXI-549 tissue probability map template. The first level fMRI models including condition of interest (mentalizing) were imported into the CONN toolbox. The data were denoised[76] including the regression of potential confounding effects characterized by white matter timeseries, CSF timeseries, SPM covariates regressors, motion parameters and their first order derivatives, ART covariates regressors, session and task effects and their first order derivatives, and linear trends within each functional run. Bandpass frequency filtering of the BOLD time series[79] was applied between 0.008 Hz and 0.09 Hz. *First-Level Inferences.* ROI-to-ROI connectivity matrices were estimated to characterize functional connectivity between each pair of the seven a priori ROIs. Functional connectivity strength was quantified using Fisher-transformed bivariate correlation coefficients from a weighted general linear model (weighted-GLM;[76]), calculated for each pair of ROIs to assess the association between their BOLD signal time series. *Second-Level Inferences.* For both child and adult groups, a weighted-GLM approach was employed[44]. A separate GLM was estimated for each individual connection, with first-level connectivity measures as dependent variables, and subjects as the independent variable, while accounting for age, sex, and site. Connection-level statistics were evaluated using multivariate parametric statistics with random effects across subjects and sample covariance estimation across multiple measurements. Inferences were made at the individual connection level, with results thresholded using familywise corrected p-FDR < 0.05[80]. Two-tailed one-sample t-tests were performed separately for the children (*n* = 80) and adults (*n* = 101) to test whether average functional connectivity between the seven regions of interest differed from zero during mentalizing, controlling for age, sex, and site. Age effects in children were plotted using a one-tailed bivariate regression analysis (based on the directed hypothesis of stronger and more extensive functional connectivity with increasing age). In adults, two two-tailed tests were conducted to account for possible in- or decreases with age in functional connectivity. To account for potential site-related variance, extracted ROI-to-ROI connectivity scores were harmonized across the two data collection sites (Basel and Zurich) using ComBat, which applies an empirical Bayes framework to model and remove unwanted site effects while preserving biological variability[81]. ComBat-harmonization was applied only on the extracted ROI-to-ROI connectivity scores and not on the whole-brain analyses. All statistical analyses included "site" as a covariate of no interest for transparency.

**Functional connectivity network strength.** An overall mentalizing network connectivity score was computed for each participant by

vectorizing the ROI-to-ROI matrices and averaging the 21 Fisher-transformed correlation scores to reflect the strength of the mentalizing network.

## Statistics and reproducibility
**Mentalizing (behavior).** A Welch *t*-test was used to compare performance between children (*n* = 79) and adults (*n* = 99).

**Effects of age.** Prior to correlation analyses, the distribution of age was examined using Shapiro-Wilk tests, which indicated deviations from normality within both the child and adult samples. In addition to whole-brain assessments, we examined the relationship between age and behavioral scores for mentalizing (n-children = 79, n-adults = 99), as well as age and neural and connectivity measures for the a priori defined ROIs for children and adults (n-children = 80, n-adults = 101) using non-parametric Spearman partial correlations, controlling for sex and site. The behavioral measure of mentalizing skills entered into these analyses was the raw number of correctly solved trials in the 20 experimental mentalizing trials (range 0-20). Percent accuracy values are reported in Table 1 for descriptive purposes only.

**Linear and non-linear age effects in children and adults.** To test life course progressions for all participants included here (e.g., in line with evidence such as[36,38,47,61,82]), significant relationships (based on the outcomes of significant effects for age and mentalizing skills (*n* = 179) and age and connectivity (*n* = 181) scores) were further explored across all and linear and quadratic models were evaluated using multiple regression.

**Moderated mediation.** To examine whether the relationship between age and mentalizing ability was mediated by long-range connectivity strength, and whether these associations varied across age group, we conducted moderated mediation analyses using PROCESS model 58[83]. Data from all participants were leveraged to explore whether the relationship between age (**X;** predictor) and mentalizing skills (**Y;** dependent variable) is mediated by changes in long-range connectivity strength (**M;** mediator) and whether these associations vary across three neurofunctional groups (**W;** moderator): children (≤ 14 years; *n* = 80), young adults (20-32 years; *n* = 27), and middle-aged adults (33-61 years; *n* = 74). The choice of these groups was both theory-driven and consistent with evidence supporting non-linear connectome maturation[36,40,47,61] as well as aligned with our curve estimations of age-related changes in connectivity strength (peaking around 32 years; Fig. 3A). The groups reflected three neurofunctional phases (increase, peak/stability, and decline) rather than strict developmental or sociological categories. A long-range connectivity strength score was calculated by averaging the connectivity strength only of those long-range connections that increased in children but decreased in adults (i.e., LTPJ-DMPFC, PC-VMPC, PC-MMPFC, and PC-DMPFC; Fig. 2, Table 5). Sex was included as a covariate of no interest in all model paths. Bootstrapping was set to 10000 samples. Raw data for this analysis have been provided in Supplementary Data.

## Reporting summary
Further information on research design is available in the Nature Portfolio Reporting Summary linked to this article.

## Data availability
All data relevant to the current manuscript are available in accordance with the restrictions specified by the corresponding ethical agreements. Due to the vulnerability of young participants, only de-identified data are shared. This may include selected behavioral data points. For neuroimaging data, extracted scores, group-level maps, and findings have been made available via NeuroVault: https://neurovault.org/collections/JQTZMGIX/. Available data points are to be obtained from the Supplementary Data file.

## Code availability

Neuroimaging analyses were conducted in SPM12[73] and CONN toolbox[76], and statistical analyses were performed in R using standard routines as described in the Methods. No custom code was required beyond these established toolboxes. R code for replicating the reported results will be made available upon request.

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

## Acknowledgements

Data collection and analysis of this research project was supported by a Jacobs Foundation Early Career Research Grant (Nr. 2016201713) and by the Swiss National Science Foundation (105314-207624). N.M.R. received additional funding from the Hochschulmedizin Zurich (HMZ, STRESS), the University of Zurich Research Priority Program 'Adaptive Brain Circuits in Development and Learning (URPP AdaBD)', and the Jacobs Foundation CRISP program.

## Author contributions

R.B. and P.D. contributed to the design, acquisition, analysis, interpretation and writing of this manuscript. D.S., E.F. and S.S. provided towards the acquisition, analysis, interpretation and writing. N.M.R. contributed to the conception, design, acquisition, analysis, interpretation and writing.

## Competing interests

The authors declare no competing interests.
