## [Transparent Peer Review file · Communications Biology]

Evolving Brain Function and Connectivity Patterns during Mentalizing in Children and Adults

Corresponding Author: Professor Nora Raschle

Version 0:

Reviewer comments:

Reviewer #1

(Remarks to the Author)

This study investigated age-related changes in the brain's mentalizing network—critical for inferring others' thoughts and intentions—by examining task-based functional connectivity using fMRI in 181 participants (80 children aged 6–14 and 101 adults aged 20–61). Results show adults outperformed children in mentalizing accuracy, with children improving with age. Connectivity strength followed a quadratic trajectory, peaking around age 32. Notably, connectivity mediated the link between age and mentalizing ability in children/adolescents but not adults. These results contribute to existing research in the field; however, there are several concerns that warrant the authors' attention.

Major concerns:

1. The introduction notes that “children typically acquire explicit mentalizing abilities between four and six years” and cites Richardson et al. (2018), which identified a mentalizing network in 3–12-year-olds. These findings collectively suggest that ages below 6 years may represent a critical period for mentalizing development. It is therefore unclear why the study's child sample (6–14 years) excludes this younger age range. Further justification for this sampling choice, as well as a discussion of it as a potential limitation, is needed.
2. Table 1 indicates that 1 child and 2 adults lacked in-scanner task performance data. A brief explanation for this missing data should be provided.
3. Participants were scanned at two Swiss sites (Basel and Zurich). While “site” was included as a covariate in statistical models (to account for potential site-specific variance), a more robust approach typically involves pre-analytic data harmonization (e.g., using ComBat), a technique designed to remove site-related variance while preserving biological variability. If such harmonization was not performed, this should be discussed as a limitation.
4. For the “long-range connectivity strength score,” connections included were those that increased in children but decreased in adults (e.g., LTPJ-DMPFC, PC-VMPC, PC-MMPFC, PC-DMPFC; Figure 2, Table 2). Notably, the “DMPFC-RSTS” connection also fits this pattern—clarification is needed on why it was excluded. Additionally, the citation to “Table 2” here should be verified; Table 5 appears to present the relevant child-adult comparative differences, suggesting this may be a typo.
5. In the “Moderated Mediation” analysis, participants were grouped as: childhood/adolescence (≤ 22 years), young adults (23–40 years), and middle-aged adults (41–61 years). Please specify the exact number of participants in each group. In Figure 3A, 7 individuals (blue dots) around 20 years of age cluster more closely with the young adult group than the childhood/adolescent group, which raises concerns about potential effects on overall trends. Exploring a more appropriate cutoff (e.g., ≤ 18 or ≤ 20 years) for the childhood/adolescence category is recommended.
6. The left TPJ and right STS show distinct age-related effects between children and adults. Discussion of why the left TPJ (vs. right) and right STS (vs. left) exhibited these age-specific effects would strengthen the manuscript. Additionally, a brief analysis of the implications of this lateralization for mentalizing function would be valuable.

Minor concerns:

1. Please ensure consistent usage of apostrophes (' and ') throughout the text. For example, on Page 3: “others' mental

states"; Page 6: "children's psychological adjustment"; Page 8: "'affective theory of mind', 'cognitive theory of mind', and 'physical causality' conditions".

2. Please ensure consistent formatting for figure citations throughout the text (e.g., use either "Figure xx" or "Fig. xx" uniformly). Additionally, verify that Figure 3A is explicitly cited in the main text.

Reviewer #2

(Remarks to the Author)

This study investigated mentalizing and its brain network in children and adults from different ages. This is an interesting study, clarifying the development of the DMN in relation with mentalizing.

I only have a few comments.

1. It is very unfortunate that the study lacks participants between 14 and 20 years old, which is a major age range for pubertal development and the development of social skills (and it is not even mentioned in the limitation section). It is further surprising that the age groups created for analyses do not account for this gaps and do 6-14, 20-40, 41-61). Based on Figure 3A, it does not feel right to include the 20 -22 years old with "children and adolescents" given the large gap. Furthermore, these are not children or adolescents anymore. A stronger justification or limitation about this is important. I also don't think that it would change the results, as the few adults individuals seem to show a similar pattern than the other young adults listed in the young adult group.

2. In the same context, From the same figure, it seems that there are only 1 or 2 individuals at 30 or 31 years old. It seems clear that the age distribution is not normal, and I am wondering how much it could bias or impact the results related to age, especially when parametric analyses have been conducted (rather than non-parametric). It seems that overall, within each group, age is not well distributed. Could the authors comment on the non-normal distribution of age within and across their groups and how / whether it could impact (or not) their results? It could potentially explain why they have limited results with age in their activation analyses as the statistical power may not be enough with the current dataset and parametric design.

3. Mentalizing (behavior): can the variables used for mentalizing be more clearly explained? Is it just the % of accuracy used for each condition? Or more than that? (the results seem to only refer to one metric, while the methods seem to refer to many related to correct, incorrect and missing trials).

4. ROIs: It should be clearly stated that the ROIs are not based on an anatomical definition, but rather a functional definition / activation. Also, it would be better if the ROIs were displayed on their own in Supplement 4 since it is not possible to see the blue for some regions (e.g., precuneus / PCC).

5. fMRI Preprocessing: "movement outliers exceeding 2mm were excluded": It is not clear what that means. Is it referring to volumes? Individuals? If volumes are excluded, did they replace the volume? If not, then the design of the task will be mismatching between individuals and that could be a problem when analyzing the group.

6. Results: "in children, mentalizing skills significantly": does that refer to the % of accuracy in the task? I think this should be more precise as I am not convinced that this accuracy measure is fully and only reflecting their "mentalizing skills" (there is also some level of attention, etc involved). Showing the plot of these findings would be good as well. Were non-parametric Spearman analyses done for these correlations with age? It is not described in the method.

7. Minor:

Several studies are cited with author names (e.g., Burnett and Blakemore, Richardson et al 2018; Sun et al 2025) but their reference number is missing.

Version 1:

Reviewer comments:

Reviewer #1

(Remarks to the Author)

The authors have addressed most of the concerns I previously raised. However, I still have a few minor points for the authors to consider:

1. In the revised manuscript, the authors applied ComBat harmonization to the connectivity scores across the two data collection sites, but not to the whole-brain analyses. Clarification is needed regarding this inconsistency.

2. For the moderated mediation results, it would be helpful to include the key statistical findings in the main text, rather than presenting all results only in the table or figure.

3. The formatting of Table 6 requires correction.

Reviewer #2

(Remarks to the Author)

Most of my concerns have been addressed.

However, I would like a new clarification about the head motion threshold. The threshold for head motion of a framewise displacement of 2 is extremely / too lenient as the recommended threshold has been typically 0.5 or even less with 0.25 (see studies by Power et al.). I am wondering whether authors have misclassified their measure by rather using the volume-to-volume values coming from the 6 motion parameters, as a threshold 2 mm or degrees have been used in the past (as is

acceptable)? It is just different from Framewise displacement and thresholds between the two metrics are very different (since they don't reflect the same). It would be good to double check and possibly add a limitation if they really used a FD of 2. In this case, I would possibly retest with a more appropriate and accepted threshold (0.5), which would correct for more volumes in their model, or add a limitation. Overall, using a FD of 2 as a threshold is not appropriate and rather not acceptable, while if they actually use actual volume-to-volume head motion (translation or rotation) this would be ok.

Reviewer #1

This study investigated age-related changes in the brain’s mentalizing network—critical for inferring others’ thoughts and intentions—by examining task-based functional connectivity using fMRI in 181 participants (80 children aged 6–14 and 101 adults aged 20–61). Results show adults outperformed children in mentalizing accuracy, with children improving with age. Connectivity strength followed a quadratic trajectory, peaking around age 32. Notably, connectivity mediated the link between age and mentalizing ability in children/adolescents but not adults. These results contribute to existing research in the field; however, there are several concerns that warrant the authors’ attention.

Thank you for the overall positive assessment and the valuable recommendations in revising our manuscript! We sincerely appreciate the time and effort taken to improve our work.

Major concerns:

1. The introduction notes that “children typically acquire explicit mentalizing abilities between four and six years” and cites Richardson et al. (2018), which identified a mentalizing network in 3–12-year-olds. These findings collectively suggest that ages below 6 years may represent a critical period for mentalizing development. It is therefore unclear why the study’s child sample (6–14 years) excludes this younger age range. Further justification for this sampling choice, as well as a discussion of it as a potential limitation, is needed.

Thank you for raising this point. We apologize that our reference placement and phrasing in the Introduction may have caused confusion regarding our chosen age range.

While implicit forms of mentalizing and their early neural precursors emerge during the first years of life (Onishi and Baillargeon, 2005; Surian et al, 2007), explicit mentalizing, which requires verbal reasoning, perspective-taking, and cognitive control, and is typically measured through explicit response-based tasks, develops more gradually throughout middle childhood and adolescence. Prior work indicates that explicit Theory of Mind tasks rely on executive and language skills that continue to mature into late childhood and adolescence (Wellman et al., 2001; Frith & Frith, 2003).

Importantly, our CAToon task assesses explicit cognitive and affective Theory of Mind, and prior validation work (Borbás et al., 2021) demonstrated that reliable task performance using such response-dependent tasks can be obtained in participants aged 5 years and older. Consequently, our recruitment focused on children aged 6–14 years,

an age range in which explicit instruction following and regulatory control are developmentally appropriate and compatible with the experimental protocol.

We have clarified this in the Introduction by adding the following statement:

“Children typically acquire explicit mentalizing abilities between four and six years, as evidenced by their ability to verbally recognize false beliefs and differentiate between their own and others’ mental states [11–13]. Although precursors of mentalizing emerge early in life, explicit and response-based forms of mentalizing that require verbal reasoning, perspective-taking, and cognitive control continue to develop throughout middle childhood and adolescence, and can be reliably assessed from approximately 5–6 years of age onward [12, 13, 16]. Accordingly, the present study targeted an age range (6–14 years) in which children can both understand task instructions and perform explicit theory of mind judgments with sufficient reliability.”

Additionally, we agree that the exclusion of younger children should be discussed as a limitation, and have added the following sentence to the Limitation section:

“As our task was designed for explicit, response-based mentalizing, children younger than 6 years were not included. Future work should adapt paradigms for younger participants to capture the emergence of early implicit and explicit mentalizing abilities during the preschool years (e.g., through more adaptive tasks; [70]).”

2. Table 1 indicates that 1 child and 2 adults lacked in-scanner task performance data. A brief explanation for this missing data should be provided.

We apologize for missing this and report the reasoning for the missing data more clearly within the main manuscript:

“In-scanner task performance data for one child and two adults were missing due to a technical malfunction of the button box. To ensure comprehension and task reliability, all participants completed an extensive training session prior to scanning, during which they verbally explained their reasoning for each trial, confirming an understanding of the task demands. Moreover, visual inspection of first-level activation maps for these participants confirmed expected task-related activation patterns, supporting the reliability of the group-level findings.”

3. Participants were scanned at two Swiss sites (Basel and Zurich). While “site” was included as a covariate in statistical models (to account for potential site-specific variance), a more robust approach typically involves pre-analytic data harmonization

(e.g., using ComBat), a technique designed to remove site-related variance while preserving biological variability. If such harmonization was not performed, this should be discussed as a limitation.

Thank you so much for this important consideration and for suggesting this analytical harmonization option using ComBat! We agree with the importance of site considerations and have employed Combat to harmonize the connectivity scores across both data collection sites. We have implemented a short description to this additional pre-analytic step in section 2.4 Neuroimaging on page 9:

“To account for potential site-related variance, extracted ROI-to-ROI connectivity scores were harmonized across the two data collection sites (Basel and Zurich) using ComBat, which applies an empirical Bayes framework to model and remove unwanted site effects while preserving biological variability [63]. All subsequent statistical analyses were conducted using the ComBat-harmonized connectivity scores, while retaining ‘site’ as a covariate of no interest for transparency..”

We have updated our results and only report analyses that were rerun with these harmonized scores. Importantly, there was no change in the significance nor the direction of the reported main results.

All data points entering group analyses (including these) are further provided in the supplement and our Limitations section was extended to include the following statement:

“Although ComBat harmonization was applied to minimize site-related variance, subtle scanner-specific effects cannot be entirely ruled out, underscoring the importance of replication in independent multi-site samples.”

4. For the “long-range connectivity strength score,” connections included were those that increased in children but decreased in adults (e.g., LTPJ-DMPFC, PC-VMPC, PC-MMPFC, PC-DMPFC; Figure 2, Table 2). Notably, the “DMPFC-RSTS” connection also fits this pattern—clarification is needed on why it was excluded. Additionally, the citation to “Table 2” here should be verified; Table 5 appears to present the relevant child-adult comparative differences, suggesting this may be a typo.

We appreciate the reviewer’s observation and apologize for not describing the selection procedure for the long-range connectivity strength score more clearly. The long-range connectivity strength score was calculated by averaging the ROI-to-ROI connectivity values that showed a significant increase in children and decrease in adults. The present analyses revealed only four such connections: LTPJ-DMPFC, PC-VMPC, PC-MMPFC, and PC-DMPFC. The DMPFC-RSTS connection, although significantly decreasing in adults, did not show a significant increase in strength in children and was therefore not included

in the composite long-range connectivity strength score. We have clarified this rationale in the Methods section and corrected the table reference from “Table 2” to the correct “Table 5”.

5. In the “Moderated Mediation” analysis, participants were grouped as: childhood/adolescence (≤ 22 years), young adults (23–40 years), and middle-aged adults (41–61 years). Please specify the exact number of participants in each group. In Figure 3A, 7 individuals (blue dots) around 20 years of age cluster more closely with the young adult group than the childhood/adolescent group, which raises concerns about potential effects on overall trends. Exploring a more appropriate cutoff (e.g., ≤ 18 or ≤ 20 years) for the childhood/adolescence category is recommended.

We thank the reviewers for their constructive feedback regarding our age group definitions in the moderated mediation analysis. Given both reviewers comments, we carefully reconsidered the age cutoffs to acknowledge the fact that the participants around 20 years of age cluster more closely with the young adults group than with the children/adolescents and the 14–20-year gap in our sample. To better reflect empirical data patterns (i.e., the observed quadratic trajectory of connectivity strength across age; Figure 3A) and theoretical models of functional brain network maturation, we revised our approach by reclassifying participants into three neurofunctional phases that correspond to distinct stages of observed network development:

- Children (≤ 14 years; $n=80$); representing the pre-adult phase characterized by rapid local connectivity increases.
- Young adults (20–32 years; $n=27$); corresponding to the ascending-to-peak phase of mentalizing network integration.
- Middle-aged adults (≥ 33 years; $n=74$); reflecting the post-peak phase of gradual connectivity decline.

This revised group distribution aligns with both our empirical data distribution (6–14 years, 20–61 years) and theoretical accounts of lifespan trajectories in large-scale network integration (e.g., Fair et al., 2008; Sun et al., 2025; Siman-Tov et al., 2017). The moderated mediation analysis was rerun with these new groups, and all main results remained consistent in direction and significance (see updated Table 6). The Methods section now lists the exact number of participants per group ($n=80$, children, $n=27$, young adults, and $n=74$ middle-aged adults).

In addition, the introduction was revised to clarify this rationale as follows:

“Finally, data across all participants will be used to explore whether the relationship between age and mentalizing skills is mediated by changes in connectivity strength and whether these associations vary for specific neurofunctional phases. We differentiate

between children (≤ 14 years), characterized by ongoing maturation and increasing functional segregation, young adults (20–32 years), representing the ascending-to-peak phase of network integration with relative stability, and middle-aged adults (33–61 years), undergoing gradual connectivity decline; in line with [36, 40, 47, 48].”

The adapted methods section reads as following:

“Moderated Mediation. *To examine whether the relationship between age and mentalizing ability was mediated by long-range connectivity strength, and whether these associations varied across age group, we conducted moderated mediation analyses using PROCESS model 58 [66]. Data from all participants were leveraged to explore whether the relationship between age (**X**; predictor) and mentalizing skills (**Y**; dependent variable) is mediated by changes in long-range connectivity strength (**M**; mediator) and whether these associations vary across three neurofunctional groups (**W**; moderator): children (≤ 14 years; $n=80$), young adults (20–32 years; $n=27$), and middle-aged adults (33–61 years; $n=74$). The choice of these groups was both theory-driven and consistent with evidence supporting non-linear connectome maturation [36, 40, 47, 64] as well as aligned with our curve estimations of age-related changes in connectivity strength (peaking around 32 years; Figure 3A). The groups reflected three neurofunctional phases (increase, peak/stability, and decline) rather than strict developmental or sociological categories.”*

6. The left TPJ and right STS show distinct age-related effects between children and adults. Discussion of why the left TPJ (vs. right) and right STS (vs. left) exhibited these age-specific effects would strengthen the manuscript. Additionally, a brief analysis of the implications of this lateralization for mentalizing function would be valuable.

Thank you for this suggestion. We have added a section to our discussion on page 22 addressing the age-related lateralization of temporoparietal and superior temporal regions.

“Notably, we observed asymmetric age-related effects within temporoparietal and superior temporal regions, consistent with prior evidence [23, 75]. Specifically, the left TPJ and right STS exhibited distinct developmental trajectories. Such hemispheric differentiation may reflect a shift from perceptually anchored to more abstract inferential processes in mentalizing. The left TPJ has been linked to conceptual perspective-taking and language-mediated reasoning about others’ beliefs, whereas the right TPJ and right STS, although not both showing age effects in the present study, are generally implicated in visual–social perception and the decoding of dynamic social cues (such as gaze direction, motion, and facial expressions; [76–78]). Developmental refinement within these regions may therefore mirror the transition from perceptual social understanding

in childhood toward increasingly abstract reasoning and linguistic forms of mental state attribution in adolescence and adulthood. This interpretation aligns with broader models of cortical specialization suggesting that functional differentiation increases during brain maturation before later convergence or reduced segregation in adulthood [40]. However, as the left STS was not included among our ROIs and whole-brain activation appeared largely bilateral, any lateralization effects should be interpreted cautiously.”

Minor concerns:

1. Please ensure consistent usage of apostrophes (' and ') throughout the text. For example, on Page 3: “others' mental states”; Page 6: “children's psychological adjustment”; Page 8: “ 'affective theory of mind', 'cognitive theory of mind', and 'physical causality' conditions”.

We thank the reviewer for noting this formatting inconsistency. We carefully reviewed the entire manuscript and ensured uniform use of typographic apostrophes (') throughout the text and within all quoted terms.

2. Please ensure consistent formatting for figure citations throughout the text (e.g., use either "Figure xx" or "Fig. xx" uniformly). Additionally, verify that Figure 3A is explicitly cited in the main text.

We standardized all figure references to a consistent format (“Figure xx”) across the manuscript and confirmed that Figure 3A is explicitly cited in the Results section.

Reviewer #2

This study investigated mentalizing and its brain network in children and adults from different ages. This is an interesting study, clarifying the development of the DMN in relation with mentalizing.

I only have a few comments.

Thanks for your helpful suggestions and careful review of our manuscript and work!

1. It is very unfortunate that the study lacks participants between 14 and 20 years old, which is a major age range for pubertal development and the development of social skills (and it is not even mentioned in the limitation section). It is further surprising that the age groups created for analyses do not account for this gaps and do 6-14, 20-40, 41-61).

Based on Figure 3A, it does not feel right to include the 20–22 years old with “children and adolescents” given the large gap. Furthermore, these are not children or adolescents anymore. A stronger justification or limitation about this is important. I also don’t think that it would change the results, as the few adults individuals seem to show a similar pattern than the other young adults listed in the young adult group.

We thank the reviewer for highlighting the absence of participants between 14 and 20 years of age and its potential implications for interpreting developmental effects. In response, and consistent with our revisions for Reviewer #1 (Point 5), we carefully reconsidered our age group definitions in the moderated mediation analysis to better align with both empirical data patterns and theoretical models of functional brain maturation. Participants were reclassified into three neurofunctional phases based on the observed quadratic trajectory of connectivity strength across the lifespan (peaking around 32 years; Figure 3A):

- Children (≤ 14 years; $n=80$); representing the pre-adult phase characterized by rapid local connectivity increases.
- Young adults (20–32 years; $n=27$); corresponding to the ascending-to-peak phase of mentalizing network integration.
- Middle-aged adults (≥ 33 years; $n=74$); reflecting the post-peak phase of gradual connectivity decline.

This revised grouping avoids artificial bridging across the 14–20 year gap, reflects the natural distribution of our sample (6–14 years, 20–61 years) and captures three functional phases of network maturation (increase, peak/stability, decline). Importantly, these data-driven boundaries are conceptually consistent with prior models of large-scale network development across the lifespan which align with our own model (e.g., Fair et al., 2008; Sun et al., 2025; Siman-Tov et al., 2017). The moderated mediation analysis was recomputed with these new cutoffs, and the main findings remained unchanged in direction and significance (see Table 6).

We have revised the description of the moderated mediation in the Methods section:

***“Moderated Mediation.** To examine whether the relationship between age and mentalizing ability was mediated by long-range connectivity strength, and whether these associations varied across age group, we conducted moderated mediation analyses using PROCESS model 58 [66]. Data from all participants were leveraged to explore whether the relationship between age (**X**; predictor) and mentalizing skills (**Y**; dependent variable) is mediated by changes in long-range connectivity strength (**M**; mediator) and whether these associations vary across three neurofunctional groups (**W**; moderator): children (≤ 14 years; $n=80$), young adults (20–32 years; $n=27$), and middle-aged adults (33–61 years; $n=74$). The choice of these groups was both theory-driven and consistent*

with evidence supporting non-linear connectome maturation [36, 40, 47, 64] as well as aligned with our curve estimations of age-related changes in connectivity strength (peaking around 32 years; Figure 3A). The groups reflected three neurofunctional phases (increase, peak/stability, and decline) rather than strict developmental or sociological categories.”

To acknowledge the remaining age gap explicitly, we added the following statement to the Limitations section.

“Additionally, our sample does not cover all developmental periods (in particular infants, toddlers, adolescents between 14 and 20 years and older adults (>61 years) are missing), which precludes drawing conclusions about continuous lifespan trajectories.”

2. In the same context, From the same figure, it seems that there are only 1 or 2 individuals at 30 or 31 years old. It seems clear that the age distribution is not normal, and I am wondering how much it could bias or impact the results related to age, especially when parametric analyses have been conducted (rather than non-parametric). It seems that overall, within each group, age is not well distributed. Could the authors comment on the non-normal distribution of age within and across their groups and how / whether it could impact (or not) their results? It could potentially explain why they have limited results with age in their activation analyses as the statistical power may not be enough with the current dataset and parametric design.

We thank the reviewer for raising this important statistical point. Shapiro-Wilk tests confirmed that age was non-normally distributed in both the children’s ($p=0.001$) and adults’ ($p<0.001$) samples. To address this, we re-ran all age-related correlation analyses using non-parametric Spearman’s rank tests instead of Pearson’s correlations. This approach is more robust to skewed data and reduces the likelihood of spurious associations. Results remained unchanged in both direction and significance, confirming the robustness of our findings.

Accordingly, we revised the methods section to specify the use of non-parametric Spearman partial correlations, controlling for sex and site.

“Prior to correlation analyses, the distribution of age was examined using Shapiro–Wilk tests, which indicated deviations from normality within both the child and adult samples. In addition to whole-brain assessments, we examined the relationship between age and behavioral scores for mentalizing, as well as age and neural and connectivity measures for the a priori defined ROIs for children and adults using non-parametric Spearman partial correlations, controlling for sex and site.”

We further note that regression analyses do not assume normality of the predictor variables but of the model residuals. We therefore examined the residual distributions for

all regression models. Residuals were non-normally distributed for the mentalizing task performance model ($p < 0.001$) but normally distributed for the average network connectivity model ($p = 0.978$). Non-normal residuals in the performance model may have inflated standard errors and reduced statistical power, particularly in underrepresented age ranges.

This is now considered by the following addition to our limitations section:

“The age distribution within our sample and age groups was non-normal, which may have inflated standard errors and reduced statistical power, limiting our ability to detect age-related effects in under-represented groups.”

3. Mentalizing (behavior): can the variables used for mentalizing be more clearly explained? Is it just the % of accuracy used for each condition? Or more than that? (the results seem to only refer to one metric, while the methods seem to refer to many related to correct, incorrect and missing trials).

We thank the reviewer for pointing out the need to clarify the behavioral metric. In Table 1 we report descriptive in-scanner performance as percent accuracy for (i) overall task performance (30 trials) and (ii) mentalizing trials (20 trials). However, for all subsequent statistical analyses (correlation, regression, moderated mediation), we used the raw number of correctly solved mentalizing trials (0–20) from the 20 experimental trials (affective and cognitive ToM). We have clarified this in the Methods and in Table 1.

“In addition to whole-brain assessments, we examined the relationship between age and behavioral scores for mentalizing, as well as age and neural and connectivity measures for the a priori defined ROIs in children and adults using non-parametric Spearman partial correlations, controlling for sex and site. The behavioral measure of mentalizing skills entered into these analyses was the raw number of correctly solved trials in the 20 experimental mentalizing trials (range 0–20). Percent accuracy values are reported in Table 1 for descriptive purposes only.”

4. ROIs: It should be clearly stated that the ROIs are not based on an anatomical definition, but rather a functional definition / activation. Also, it would be better if the ROIs were displayed on their own in Supplement 4 since it is not possible to see the blue for some regions (e.g., precuneus / PCC).

We thank the reviewer for pointing out that our Regions of Interest (ROIs) required clearer description and improved visualization. We now specify that our ROIs were functionally defined, rather than anatomically delineated. Specifically, ROIs were based on functional activation maps from an independent group analysis of 462 neurotypical adults ([53]), representing canonical nodes of the mentalizing network.

To enhance visualization, we added a new supplementary figure (Supplement 3), ensuring that regions such as precuneus/PCC are clearly visible.

The revised Methods text now reads:

“Regions of Interest (ROIs). ROIs were functionally defined based on activation maps from an independent group analysis of 462 neurotypical adults [54] and comprised bilateral TPJ [L/RTPJ], PC, right STS [RSTS], dorsomedial PFC [DMPFC], middle medial PFC [MMPFC], and ventromedial PFC [VMPFC]. Anatomical labels are provided for orientation only. All ROIs are visualized in **Supplement 3.**”

5. fMRI Preprocessing: “movement outliers exceeding 2mm were excluded”: It is not clear what that means. Is it referring to volumes? Individuals? If volumes are excluded, did they replace the volume? If not, then the design of the task will be mismatching between individuals and that could be a problem when analyzing the group.

We thank the reviewer for pointing out the need to clarify our motion-correction procedure. The statement “movement outliers exceeding 2 mm were excluded,” refers to individual fMRI volumes (TRs), not participants. For each run, framewise displacement was computed from the realignment parameters, and any volumes showing displacement > 2 mm was labelled as a motion outlier. These outlier volumes were modelled as single-impulse nuisance (“spike”) regressors in the first-level GLM (i.e., censored rather than deleted or interpolated), following standard practice in CONN/SPM. The six standard motion parameters were also included as nuisance covariates.

Our fMRI task employed a block design (19 s per block, corresponding to ~8–9 volumes) with 10 trials per condition. The proportion of censored volumes per participant ranged from 0% to 4.6% (mean: children 1.05, adults 0.74 volumes). Because the fraction of censored volumes was small and distributed across blocks, design efficiency and first-level beta estimates can be assumed to be minimally affected. Importantly, first-level beta estimates are computed only from the retained time points, and participants can have slightly different numbers of valid TRs without biasing group-level contrasts (only within-subject variance changes, which is carried forward).

We confirmed that no task block contained extensive censoring, no runs were excluded due to motion, and the proportion of censored volumes did not differ between groups ($p = 0.604$; see Supplement 4).

To improve clarity, we have also revised our Methods section accordingly:

“Movement and signal outliers were detected using the ART toolbox [55]. Framewise volume-to-volume displacement was computed for each run and volumes with displacement > 2 mm were modeled with single-impulse nuisance (“spike”) regressors in the first-level GLM, rather than deleted or temporally interpolated. Six movement and

one aggregated outlier regressor were added in each participant's first-level model. Given the block design and low fraction of censored volumes, this approach was expected to have negligible impact on model efficiency."

6. Results: "in children, mentalizing skills significantly": does that refer to the % of accuracy in the task? I think this should be more precise as I am not convinced that this accuracy measure is fully and only reflecting their "mentalizing skills" (there is also some level of attention, etc involved). Showing the plot of these findings would be good as well. Were non-parametric Spearman analyses done for these correlations with age? It is not described in the method.

The term "mentalizing skills" refers to the number of correctly solved trials in the mentalizing condition, as now clarified in the Methods section on page 10. We acknowledge that this accuracy measure reflects overall task performance, which includes attentional and cognitive control components in addition to mentalizing per se, and have adjusted the wording throughout the Methods, Results and Discussion to reflect this more accurately.

Given the non-normal age distribution in children and adults, we re-ran all age-related correlation analyses using non-parametric Spearman's rank tests, including the association between age and mentalizing performance, as well as correlations with functional activation and network connectivity. The direction and significance of results were unchanged, and the updated coefficients are now reported in the Results section on pages 11 and 14.

To address the reviewer's request for visualization, we added a supplementary figure (new **Supplement 5**) illustrating the relationship between age and mentalizing accuracy for the children's group. We also revised the Methods section to read:

"In addition to whole-brain assessments, we examined the relationship between age and behavioral scores for mentalizing, as well as age and neural and connectivity measures for the a priori defined ROIs for children and adults using non-parametric Spearman partial correlations, controlling for sex and site. The behavioral measure of mentalizing skills entered into these analyses was the raw number of correctly solved trials in the 20 experimental mentalizing trials (range 0–20). Percent accuracy values are reported in Table 1 for descriptive purposes only."

7. Minor: Several studies are cited with author names (e.g., Burnett and Blakemore, Richardson et al 2018; Sun et al 2025) but their reference number is missing.

We thank the reviewer for noticing this oversight. We have carefully reviewed all in-text citations and added the corresponding reference numbers to ensure full consistency with the numbered reference style used throughout the manuscript.

Reviewer #1:

The authors have addressed most of the concerns I previously raised. However, I still have a few minor points for the authors to consider:

Thank you for acknowledging our efforts to incorporate your suggestions for improvement.

1. In the revised manuscript, the authors applied ComBat harmonization to the connectivity scores across the two data collection sites, but not to the whole-brain analyses. Clarification is needed regarding this inconsistency.

We thank the reviewer for raising this point. We apologize that our rationale was not sufficiently explicit. ComBat harmonization was applied only to the ROI-to-ROI connectivity scores because these analyses relied on subject-level summary measures extracted from predefined ROIs. ComBat is well suited for harmonizing such vectorized measures, where scanner/site variance can be isolated and removed while preserving biological variance.

In contrast, ComBat is not typically applied to voxel-wise first-level fMRI time series or whole-brain statistical maps because (i) voxel-wise ComBat may distort spatial autocorrelation structure, (ii) preprocessing pipelines already addressed scanner differences through identical acquisition protocols (except for hardware differences), normalization, smoothing, and explicit inclusion of site as a covariate, and (iii) whole-brain analyses rely on mass-univariate modeling where site is modeled as a nuisance regressor.

Thus, our approach follows standard practice in multi-site task fMRI studies:

- whole-brain analyses: site included as covariate, no ComBat on images;
- ROI-to-ROI connectivity: ComBat applied to extracted connectivity summaries.

We added a brief clarification to the Methods to make this explicit.

“ComBat-harmonization was applied only on the extracted ROI-to-ROI connectivity scores and not on the whole-brain analyses. All statistical analyses included “site” as a covariate of no interest for transparency.”

2. For the moderated mediation results, it would be helpful to include the key statistical findings in the main text, rather than presenting all results only in the table or figure.

Thank you for this helpful suggestion. We have now added a concise summary of the main moderated mediation findings directly in the Results section. We have reported all main effects and interactions and have highlighted (i) the significant indirect effect of age via long-range connectivity in children, (ii) the absence of

significant indirect effects in young and middle-aged adults. This addition improves readability and ensures the most important statistics are immediately visible in the main narrative.

3. The formatting of Table 6 requires correction.

We thank the reviewer for noticing this. We corrected Table 6 formatting in the revised manuscript by aligning the columns, ensuring consistent decimal precision in line with APA 7 guidelines.

Reviewer #2:

Most of my concerns have been addressed.

Thank you for recognizing the work we have put into integrating your feedback.

However, I would like a new clarification about the head motion threshold. The threshold for head motion of a framewise displacement of 2 is extremely / too lenient as the recommended threshold has been typically 0.5 or even less with 0.25 (see studies by Power et al.). I am wondering whether authors have misclassified their measure by rather using the volume-to-volume values coming from the 6 motion parameters, as a threshold 2 mm or degrees have been used in the past (as is acceptable)? It is just different from Framewise displacement and thresholds between the two metrics are very different (since they don't reflect the same). It would be good to double check and possibly add a limitation if they really used a FD of 2. In this case, I would possibly retest with a more appropriate and accepted threshold (0.5), which would correct for more volumes in their model, or add a limitation. Overall, using a FD of 2 as a threshold is not appropriate and rather not acceptable, while if they actually use actual volume-to-volume head motion (translation or rotation) this would be ok.

We thank the reviewer for this important clarification and apologize for the confusion created by using the incorrect terminology. We agree that a framewise displacement (FD) threshold of 2 mm would indeed be inappropriate and far above commonly recommended values (e.g., 0.25–0.5 mm). We did not use Power-style FD in our preprocessing (Power, 2012, suggesting an aggregate summary). Instead, we used the volume-to-volume displacement metric provided by the ART toolbox, which is based directly on the realignment parameters for translation. In this context, a 2 mm threshold refers to raw translation differences between consecutive volumes, not to FD. Importantly, this threshold is consistent with the default, historically used practice in ART/SPM and in line with procedures from our previous publications for detecting motion outliers.

We have corrected the terminology in the Methods section of the manuscript and now explicitly state that we used “volume-to-volume displacement derived from the six from realignment parameters” rather than “framewise displacement”, since no FD-based thresholding was applied.

We thank the reviewer for catching this imprecision; the revised text should now accurately reflect the procedure used.

*“Movement and signal outliers were detected using the ART toolbox [56]. ART identifies outlier volumes based on volume-to-volume displacement derived from the six realignment parameters. Volumes showing translational change exceeding 2 mm or global intensity outliers were modeled with single-impulse nuisance (‘spike’) regressors in the first-level GLM, rather than deleted or temporally interpolated (see **Supplement 4**). Six motion parameters and one aggregated outlier regressor were added in each participant’s first-level model.”*